# SpecGaussian with latent features: A high-quality modeling of the view-dependent appearance for 3D Gaussian Splatting

## ABSTRACT

Recently, the 3D Gaussian Splatting (3D-GS) method has achieved great success in novel view synthesis, providing real-time rendering while ensuring high-quality rendering results. However, this method faces challenges in modeling specular reflections and handling anisotropic appearance components, especially in dealing with view-dependent color under complex lighting conditions. Additionally, 3D-GS uses spherical harmonic to learn the color representation, which has limited ability to represent complex scenes. To overcome these challenges, we introduce Lantent-SpecGS, an approach that utilizes a universal latent neural descriptor within each 3D Gaussian. This enables a more effective representation of 3D feature fields, including appearance and geometry. Moreover, two parallel CNNs are designed to decoder the splatting feature maps into diffuse color and specular color separately. A mask that depends on the viewpoint is learned to merge these two colors, resulting in the final rendered image. Experimental results demonstrate that our method obtains competitive performance in novel view synthesis and extends the ability of 3D-GS to handle intricate scenarios with specular reflections.

## CCS CONCEPTS

• **Computing methodologies** → **Reflectance modeling**; *Rasterization*; *Visibility*; • **Networks** → *Network algorithms*.

## KEYWORDS

neural rendering, computer graphics, novel view synthesis, deep learning

## 1 INTRODUCTION

The 3D reconstruction and novel view synthesis represent formidable challenges in the fields of computer vision and computer graphics, impacting a wide range of applications including augmented and virtual reality (AR/VR), autonomous navigation, and 3D content creation. Traditional methods, relying on primitive representations like meshes and points[4, 23, 34], often compromise quality to achieve faster rendering. However, recent advancements, particularly in Neural Radiance Fields (NeRF)[1, 8, 21, 22], have made significant progress in this task by implicitly representing the scene's geometry and radiance information, thus achieving

*ACM MM, 2024, Melbourne, Australia*

© 2024 Copyright held by the owner/author(s). Publication rights licensed to ACM.
ACM ISBN 978-x-xxxx-xxxx-x/YY/MM
https://doi.org/10.1145/nnnnnnn.nnnnnnn

high-fidelity visual outputs. The primary drawback of NeRF, however, lies in its computationally intensive volume rendering process, which hampers its suitability for real-time rendering applications.

A recent and promising alternative to implicit radiance field representations is 3D Gaussian Splatting (3D-GS)[10], which has emerged as a viable solution for real-time rendering, delivering state-of-the-art quality results. 3D-GS leverages a collection of 3D Gaussian primitives to explicitly model the scene, incorporating a differentiable rasterization pipeline that circumvents the time-comsuming ray sampling process. Moreover, clone and split strategies are employed to ensure that 3D Gaussians, initialized from Structure-from-Motion (SfM), provide better coverage of the entire scene.

Despite its exceptional performance, 3D-GS encounters challenge when it comes to accurately capturing the complex and pronounced specular effects observed in specific scenes. This limitation arises from the inherent constraints of low-order spherical harmonics (SH) used in 3D-GS, as they are adept at modeling only subtle view-dependent phenomena. A recent study, referred to as Spec-Gaussian[33], integrates an Anisotropic Spherical Gaussian (ASG) technique to overcome this limitation. However, retaining SH coefficients for each Gaussian is not only unnecessary but also potentially leads to excessive memory usage. Moreover, it does not resolve the issue of significant artifacts that frequently occur within 3D-GS[36].

In order to address these issues, we propose Lantent-SpecGS, which innovatively integrates a versatile latent neural descriptor within each 3D Gaussian. This descriptor allows each Gaussian to encapsulate critical scene attributes such as local geometry, color, and material properties. By embedding these latent neural descriptors, our method not only enhances the generality and applicability of 3D-GS but also refines the rendering of view-dependent effects. Specifically, we utilize these latent descriptors to compute the normal directions associated with each Gaussian, subsequently employing these normal vectors to obtain a view-mask feature. Our experiments indicate that this view-mask feature significantly influences the decoding of view-dependent colors, thereby overcoming some of the limitations observed in traditional 3D-GS approaches.

After employing Feature Gaussian Splatting, our approach yields a diffuse feature map, a view-dependent feature map, and a view-mask map. For color decoding, we adopt the Cook-Torrance model[6] to decompose the color into diffuse color and view-dependent colors for individual decoding processes. The diffuse color is decoded by our Diffuse-UNet, which enhances the smoothness of this component. Additionally, The UNet architecture also assists in addressing scenarios with sparse viewpoints, improving rendering quality by mitigating issues such as jagged edges and pixel anomalies. This decomposition technique allows us to represent colors as a superposition of physically interpretable components, thereby enhancing

realism and fidelity of the rendered colors. Moreover, for the decoding of view-dependent color, our method creates view-mask that captures spatial lighting variations across different viewpoints. We also incorporate view embeddings to simulate view-dependent effects under near-field lighting conditions. By multiplying the view-mask map with the view-dependent color map, we obtain the final view-dependent color output. This integration can be conceptualized as an attention mechanism, focusing on complex lighting conditions and delivering more precise rendering results. We have rigorously tested our method on several public datasets such as Shiny Dataset, and compare it with existing state-of-the-art methods. The experimental results show that our method significantly improves the performance of specular reflection modeling and view-dependent color synthesis. When synthesizing novel views, our method exhibits superior rendering quality to existing methods.

In summary, the main contributions of our approach are as follows:

- A novel 3D Gaussian splatting-inspired framework is designed by embedding latent feature in each 3D Gaussian, which significantly boosts the representation capacity of 3D-GS, allowing for the unified incorporation of both geometric and appearance information.
- An efficient rendering paradigm by cooperating two parallel CNNs to decoder diffuse and specular colors separately, which can improve the rendering quality in real-world scenes, especially for specular reflections.
- Our full pipeline achieves state-of-the-art novel view synthesis performance in scenes with specular highlights, as evidenced by our results on Shinny datasets.

## 2 RELATED WORK

### 2.1 3D Scene Representations for NVS

Novel view synthesis (NVS) involves generating new images from viewpoints that are distinct from the original captures. Recently, Neural Radiance Field (NeRF)[21] has garnered significant attention due to its impressive performance in NVS. The vanilla NeRF adopts coordinate-based MLPs to implicitly represent both the geometry and the radiance information of a scene, rendering images through the volume rendering technique. In subsequent research, the primary focus in this field has been on enhancing either the quality[1–3, 29] or efficiency[8, 17, 22, 35] of rendering. However, achieving both simultaneously in NeRF-based methods has proven to be challenging. Further research has broadened the usefulness of NeRF to various applications, including mesh reconstruction[15, 18, 37], inverse rendering[26, 30, 32, 39, 40], autonomous driving[5, 14, 41] and video generation[13, 25, 31].

In contrast to the NeRF-style implicit scene representation, 3D Gaussian Splatting (3D-GS)[10] adopts an explicit approach by representing scenes using a collection of anisotropic 3D Gaussians, which are initialized from structure from motion (SfM). Additionally, 3D-GS introduces a differentiable tile-based rasterizer that allows for real-time rendering without compromising on high quality. Variants of 3D-GS are dedicated to addressing challenges such as the heavy storage requirement[7, 12, 24] and the occurrence of artifacts under certain conditions[16, 36]. Despite the impressive outcomes, its approach of associating each Gaussian with spherical harmonics

(SH) parameters leads to substantial memory consumption and and fails to capture and reproduce the appearance of specular surfaces with precision.

### 2.2 View-Dependent Modeling

NeRF leverages the positional encoding[27] of the input ray direction to account for view-dependency, while 3D-GS represents color variations from different viewpoints using spherical harmonics (SH). Nonetheless, both are generally constrained to modeling only slight view-dependent phenomena. Advances have been made in both methods to enhance the reconstruction of scenes with stonger specular reflections, thereby extending their utility to more realistic settings. Ref-NeRF[28] refines NeRF's handling of glossy scenes by reparameterizing the outgoing radiance using the reflection about the local surface normal instead of ray direction, introducing an Integrated Directional Encoding and decomposing radiance into diffuse and specular components to for better material and texture variation management. SpecNeRF[20] presents a novel Gaussian directional encoding to effectively model specular reflections under spatially varying near-field lighting conditions. This work employs a data-driven normal estimation approach early in training to improve surface geometry reconstruction, leading to more realistic and accurate rendering of glossy surfaces in 3D scenes.

Even so, they still face challenges in achieving real-time rendering due to the implicit nature of NeRF's representation, which requires repeatedly querying a neural network throughout the rendering process. Therefore, Spec-Gaussian[33] advances 3D-GS by utilizing an Anisotropic Spherical Gaussian (ASG) appearance field in place of traditional spherical harmonics. This innovation permits the capture of high-frequency view-dependent appearance information without increasing the number of 3D Gaussians. Furthermore, this method incorporates anchor-based, geometry-aware 3D Gaussians for efficiency inpired by [19] and a coarse-to-fine training strategy to eliminate artifacts, thus achieving exceptional depiction of specular highlights. However, retaining SH coefficients for each Gaussian is redundant and leads to excessive memory usage. Moreover, these aforementioned methods often struggle with sparse viewpoints, such as scene edges, and are prone to producing artifacts. To address these issues, our approach discards all SH parameters and instead leverages latent neural descriptor composed of diffuse and view-dependent features. We then generate a diffuse feature map via $\alpha$-blending and decode it into base color and view-dependent color based on the dichromatic reflection model for each Gaussian. The UNet-style diffuse decoder used in our approach effectively eradicates artifacts and ensures the smoothness of synthesized images. Experiments reveal that our method not only shows notable improvements in reconstruction metrics but also yields renderings with strong physical interpretability in various specular scenarios.

## 3 METHODS

The overview of our method is illustrated in Figure 1. Starting with a series of images taken from different camera poses of a static scene, we first utilize Colmap to generate a sparse point cloud via Structure-from-Motion (SfM). Following this, we initialize 3D Gaussians, each endowed with neural descriptors that encompass

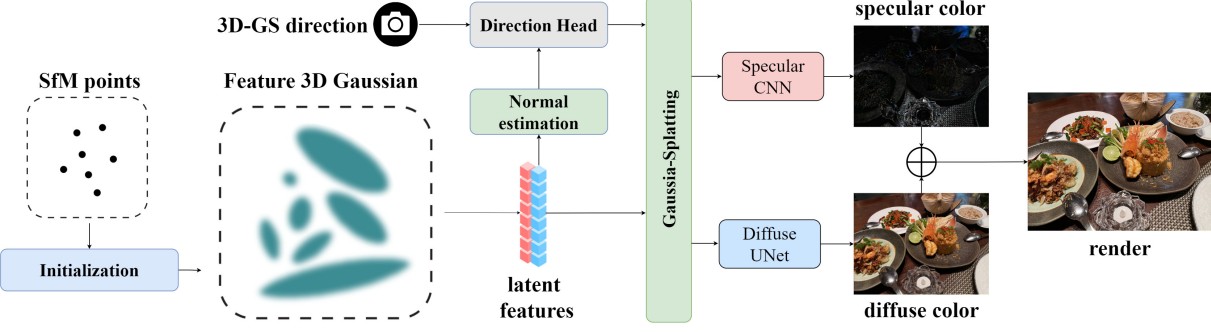

**Figure 1: Pipeline of our proposed Latent-SpecGaussian. It consists of three steps: First, initializing with SfM points derived from COLMAP. Second, optimizing latent feature within each 3d gaussian and directional feature. Finally, splatting to generate multiple feature maps and decoding color using two parallel networks.**

diffuse features, view-dependant features, a view-mask feature, and a normal vector. These four components are concatenated and subsequently processed through Feature Gaussian Splatting, leading to the generation of several feature maps: a diffuse feature map, a view-dependent feature map, and a view-mask map. For the diffuse feature map and the view-dependant feature map, we employ an UNet-style network and a compact CNN network with embedded viewpoint encoding to decode the diffuse color and the view-dependant color respectively. Drawing inspiration from the Cook-Torrance model[6], the view-dependent color image is multiplied by the view-mask map. This product is then overlaid onto the diffuse color image in the synthesis of the final novel view output.

## 3.1 Preliminaries

### 3.1.1 *3D Gaussian Splatting*.
3D-GS[10] explicitly represents the scene with a set of anisotropic 3D Gaussians that have optimizable attibutes including 3D position, opacity, anisotropic covariance, and spherical harmonic (SH) coefficients. These primitives are subsequently splatted into images via a tile-based rasterizer.

Specifically, the $i$-th 3D Gaussian primitive $\mathcal{G}_i$ is parameterized by an opacity $\alpha$, a center $\mu$, and a full 3D covariance matrix $\Sigma$ defined in world space:

$$\mathcal{G}_i(x) = e^{-\frac{1}{2}(x-\mu_i)^T \Sigma_i^{-1}(x-\mu_i)} \tag{1}$$

where $x$ is an arbitrary position within the 3D scene. The covariance matrix $\Sigma$, which is positive semi-definite, is derived from a scaling matrix $S$ and a rotation matrix $R$, expressed as $\Sigma = RSS^T R^T$. Afterwards, $\Sigma$ is transformed to $\Sigma'$ in camera coordinates when 3D Gaussians $\mathcal{G}$ are projected to 2D Gaussians $\mathcal{G}'$ for rendering:

$$\Sigma' = JW\Sigma W^T J^T \tag{2}$$

where $W$ is the extrinsic matrix and $J$ is the Jacobian of the affine approximation of the projective transformation. Then a tile-based rasterizer is designed to efficiently sort the 2D Gaussians according to depths and employ $\alpha$-blending to compute the color $C$ at pixel $x'$:

$$C(x') = \sum_{i=1}^{N} c_i \alpha_i \mathcal{G}_i(x) \prod_{j=1}^{i-1}(1 - \alpha_j \mathcal{G}'_j(x')) \tag{3}$$

where $x'$ is the queried pixel, $N$ stands for the number of ordered 2D Gaussians associated with that pixel, and $c_i$ is the decoded color of $i$-th Gaussian $\mathcal{G}_i$ from its coefficients of spherical harmonics (SH). During optimization, 3D Gaussians are adaptively added and occasionally removed for precise representation of the scene. We refer the reader to [10] for details.

### 3.1.2 *Cook-Torrance*.
Compared to NeRF, Ref-NeRF[28] decomposes outgoing radiance into diffuse color $c_d$ and specular color $c_s$, and then combines them to obtain a single color value:

$$c = \gamma(c_d + s \odot c_s) \tag{4}$$

where $\odot$ denotes dot product, and $\gamma$ is a fixed tone mapping function. And $s$ represents the specular tint predicted by a spatial MLP similar to $c_d$, while $c_s$ is predicted by a directional MLP.

This decomposition aligns with the principles posited in the Cook–Torrance approximation[6] of the rendering equation. Within this context, the expression $s \odot c_s$ embodies the split-sum approximation of the specular component of the Cook-Torrance model. Specifically, $c_s$ is correlated with the preconvolved incident light reflecting off the surface, while $s$ approximates the pre-integrated bidirectional reflectance distribution function (BRDF).

## 3.2 Latent 3D-GS and Feature Gaussian Splatting

### 3.2.1 *Latent 3D-GS*.
In contrast to the original 3D-GS, our method introduces several key modifications to enhance color expression. Notably, we have replaced the spherical harmonic (SH) parameters adhering to each 3D Gaussian with more versatile and optimized neural descriptors. This substitution facilitates the learning of latent representations encapsulating local geometry, color, and material properties within a scene, thereby achieving a more detailed and accurate depiction.

In our method, the $i$-th Gaussian carries a 16-dimensional latent feature vector $f_i$, which is bifurcated into two distinct components: an 8-dimensional vector of diffuse latent features $f_d$ and an 8-dimensional vector of specular latent features $f_s$. These components are structured as follows:

$$f_i = f_d \oplus f_s \tag{5}$$

where ⊕ denotes the operation of concatenation. And the latent features are randomly initialized, akin to other base attributes such as scales and rotations associated with each Gaussian. These latent features are further decoded by specific networks to extract color components related to diffuse reflection and specular reflection, respectively.

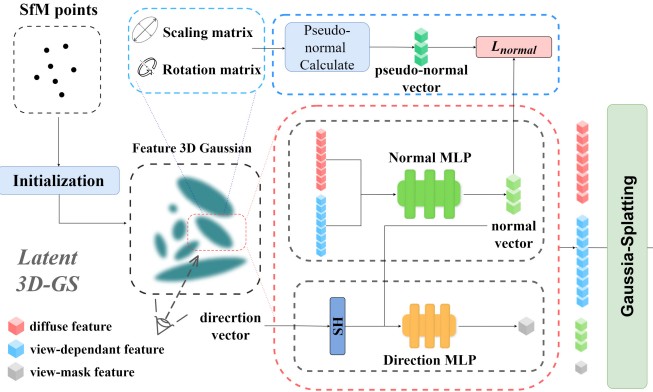

**Figure 2: Schematic of Latent 3D-GS. In addition to the latent features attached to the 3DGS, we also use these features to predict normals and decode the viewpoint mask features.**

### 3.2.2 View-Mask Feature and Normal Estimation.

The accurate understanding and representation of the Bidirectional Reflectance Distribution Function (BRDF) are essential when using 3D Gaussian splatting for photorealistic rendering of intricate scenes. The BRDF describes the material properties of surfaces, including characteristics such as reflectance, glossiness, and transparency. By faithfully modeling the BRDF, we enhance the realism and accuracy of rendered images, particularly in complex scenes.

In our approach, we assign a view-mask feature to each latent 3D-GS in order to learn the distribution of the BRDF function. For each latent 3D-GS, we have attributes including its position, opacity, rotation matrix, and scaling matrix. To determine the viewpoint direction vector for each latent 3D-GS, we compute the vector difference between its position and the camera center. Additionally, we use a small MLP to predict the normal direction, as described by the following equation:

$$\Psi(f_s, f_d) = \mathrm{n} \tag{6}$$

Considering the BRDF function, the equation for outgoing light $L_o$ in a given direction $\boldsymbol{\omega_o}$ at a point $\boldsymbol{x}$ is described by:

$$L_o\left(\boldsymbol{\omega_o}, \boldsymbol{x}\right) = \int_\Omega f\left(\boldsymbol{\omega_o}, \boldsymbol{\omega_i}, \boldsymbol{x}\right) L_i\left(\boldsymbol{\omega_i}, \boldsymbol{x}\right)\left(\boldsymbol{\omega_i} \cdot \boldsymbol{n}\right) d\boldsymbol{\omega_i}, \tag{7}$$

where $L_i$ corresponds to the incident light radiance coming from direction $\boldsymbol{\omega_i}$, and $f$ represents the point's BRDF properties. The integration domain is the upper hemisphere $\int_\Omega$ defined by the point $\boldsymbol{x}$ and its normal $n$. And $\boldsymbol{\omega_o}$ can be expressed as:

$$\omega_o = 2\left(\omega_i \cdot n\right) \cdot n - \omega_i, \tag{8}$$

where $\boldsymbol{\omega_i}$ is the incident light direction that can be approximated by the opposite direction of the viewing direction, and $n$ is the approximate surface normal.

For each Gaussian, we utilize a small MLP to learn the view-mask feature $f_m$. Specifically, the input to this MLP comprises the predicted normal direction $\mathbf{n}$ and the view direction $\mathbf{d}$, the latter of which is transformed using Spherical Harmonic encoding. Equation 9 represents this process:

$$f_m = F_{\boldsymbol{\theta}}\left(\lambda_{\mathrm{SH}}(\mathbf{d}), \mathrm{n}\right), \tag{9}$$

where $\lambda_{\mathrm{SH}}(\cdot)$ represents SH encoding. Directly learning the normal vector from latent features can result in significant errors. To mitigate these inaccuracies, we introduces a pseudo normal calculation as regularization. Typically, the accurate estimation of normal vectors requires a continuous surface; however, the discrete nature of scene representations poses a substantial challenge to this direct estimation. Empirical observations indicate that configuration of 3D-GS tends to approximate a flatter geometry as the optimization process. Based on this phenomenon, it becomes pragmatically feasible to approximate the normal vector by identifying the shortest axis of the Gaussian. This approach aligns with Spec-Gaussian [33], where the shortest axis, determined by the rotation and size matrices of each Gaussian, is used as the pseudo normal direction. And we also flip the normal direction based on the current view direction. This process can be represented by the following formula:

$$\mathbf{n} = \begin{cases} -\boldsymbol{n}, & \boldsymbol{n} \cdot \boldsymbol{v} < 0 \\ \boldsymbol{n}, & \boldsymbol{n} \cdot \boldsymbol{v} \ge 0 \end{cases} \tag{10}$$

While the pseudo normals estimated through this method may not perfectly align with real-world physical normals, employing them as supervisory signals effectively enhances the network's ability to approximate actual normal directions and precisely capture reflection distributions, essential for sophisticated view-dependent learning.

### 3.2.3 Feature Gaussian Splatting.

Rather than splatting colors, we now splat the features due to the replacement of SH coefficients with neural descriptors. Following the the normal estimation and view-mask feature prediction methods aforementioned in Section 3.2, we concatenate the diffuse latent feature and the specular latent features on latent 3D-GS, along with the view-mask feature and the estimated normal. These concatenated features are subsequently used in Feature Gaussian Splatting to generate latent feature maps, which are then decoded into colors as detailed in Section 3.3.

Parallel to the rasterization pipeline of 3D-GS, Feature Gaussian Splatting isimilarly relies on $\alpha$-blending approach. Specifically, at each pixel on the image plane, the features are accumulated with weights determined by the alpha values of each Gaussian, as described by the following equation:

$$f(\mathbf{p}) = \sum_{i \in N} T_i \alpha_i f_i, \quad \alpha_i = \sigma_i e^{-\frac{1}{2}(\mathbf{p}-\mu_i)^T \Sigma'(\mathbf{p}-\mu_i)}, \tag{11}$$

where $\mathbf{p}$ is the pixel coordinates, $T_i$ is the transmittance defined by $\Pi_{j=1}^{i-1}\left(1 - \alpha_j\right)$, $f_i$ denotes the latent feature of the sorted Gaussian distribution associated with the query pixel, and $mu_i$ denotes the coordinates when projected onto a 3D Gaussian to a 2D image plane.

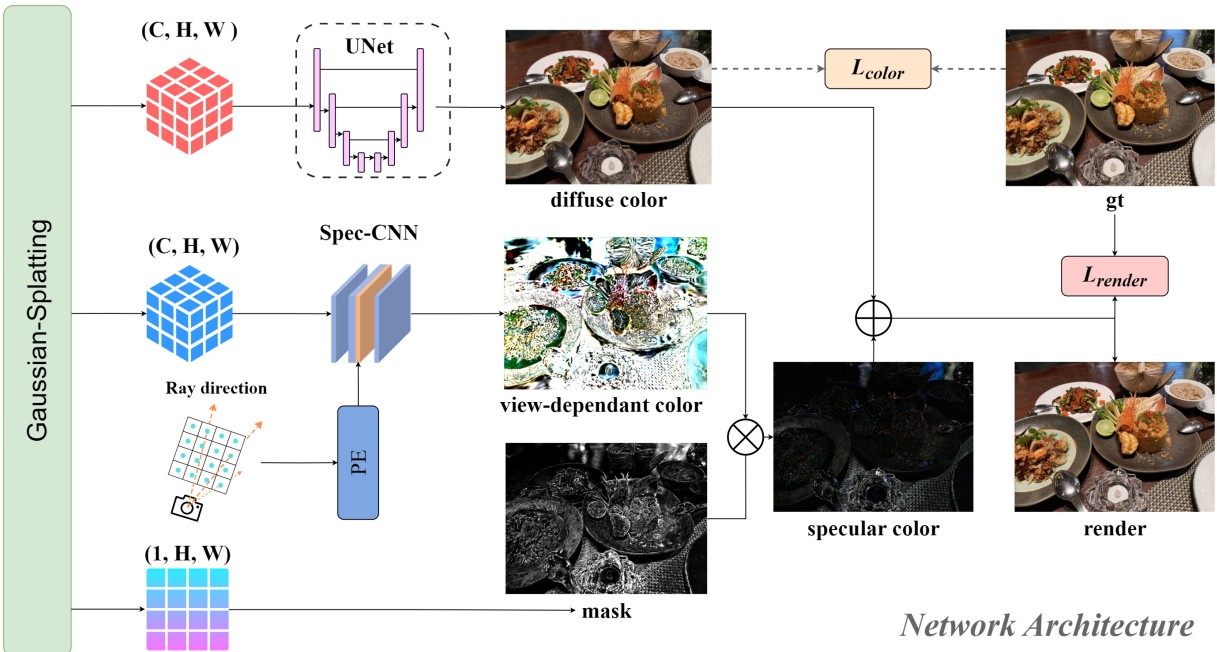

**Figure 3: Network Architecture. It includes a UNet to decode diffuse color and a CNN for decoding view-dependent color. The view-dependent color is multiplied by the view mask to superimpose the diffuse colors, resulting in the final rendering.**

## 3.3 Decoder Architecture

Our decoder architecture is depicted in Figure 3. To effectively decompose the diffuse and specular components of color, we have developed two specialized networks. The first, a UNet-style network, is tasked with decoding the diffuse feature map, while the second, a CNN-based network with viewpoint embeddings, is designed for decoding the view-dependent feature map. In the subsequent text, we simply called them as Diffuse-UNet and Specular-CNN.

The original 3D-GS method densifies sparse point clouds through splitting and pruning strategies, but this approach may neglect the scene's underlying structure, leading to redundant Gaussian distributions. These excess distributions can produce jagged edges or anomalous Gaussians, diminishing the method's robustness to new viewpoints. Furthermore, 3D-GS struggles in scenarios with sparse viewpoints, such as scene edges or the initial frames of a sequence, often resulting in noticeable blurriness and artifacts.

To solve these problems, we introduce the Diffuse-UNet. This network efficiently down-samples the diffuse feature map to one-eighth of its original size, and then employs up-sampling to restore it. Residual modules bolster feature extraction at each convolution step, complemented by skip connections for enhanced feature fusion. The base layer uses $3 \times 3$ convolutions with ELU activation, and LayerNorm is applied in the residual modules to adapt to single-batch optimizations in the UNet framework. The integration of Diffuse-UNet into the 3D-GS framework is designed to substantially improve reconstruction quality and minimize the occurrence of artifacts. Notably, Diffuse-UNet helps in filling the gaps and smoothing out jagged edges in synthesized images, especially in areas with sparse viewpoints. The strategic use of residual modules

and skip connections further refines the image generation process, significantly elevating the overall visual quality.

For the Specular-CNN network designed to decode the specular feature map, we begin with a base convolutional layer to expand the dimensions of the view-dependent feature map. Following the viewpoint embedding technique used in NeRF, we calculate a ray direction for each pixel in the 2D image, encode these directions with positional encoding, and concatenate them with the expanded specular feature map. We then decode the specular colors using the enriched, concatenated features. This approach enhances the rendering of view-dependent effects in the image.

Based on the Cook-Torrance physical model[6], our method multiplies the specular color with the view-mask map and then adds it to the diffuse color to obtain the final result. The total process can be expressed as:

$$c = F_{\theta_{UNet}}(\mathbf{f}_d) + \left( F_{\theta_{Specular-CNN}} \left( \lambda_{\mathrm{PE}}(\mathbf{d_{pixel}}), \mathbf{f}_s \right) \cdot mask \right) \quad (12)$$

## 3.4 Loss Function

Our loss function is designed as :

$$L = L_{\mathrm{renders}} + \lambda_{\mathrm{diffuse}} L_{\mathrm{diffuse}} + \lambda_{\mathrm{normal}} L_{\mathrm{normal}}, \quad (13)$$

where,

$$L_{\mathrm{renders}} = (1 - \lambda_{\mathrm{D\text{-}SSIM}}) L1(renders, gt) \\ + \lambda_{\mathrm{D\text{-}SSIM}} L_{\mathrm{D\text{-}SSIM}}(renders, gt), \quad (14)$$

$$L_{\mathrm{diffuse}} = (1 - \lambda_{\mathrm{D\text{-}SSIM}}) L1(diffuse, gt) \\ + \lambda_{\mathrm{D\text{-}SSIM}} L_{\mathrm{D\text{-}SSIM}}(diffuse, gt), \quad (15)$$

$$L_{\text{normal}} = 1 - \frac{\mathbf{n} \cdot \mathbf{n}_{pseudo}}{\|\mathbf{n}\| \cdot \|\mathbf{n}_{pseudo}\|} \tag{16}$$

In our method, $L_{\text{renders}}$ and $L_{\text{diffuse}}$ mirror the loss functions from the original 3D-GS. $L_{\text{renders}}$ is designed to compare the final rendered image against the ground truth, assessing overall synthesis quality, while $L_{\text{diffuse}}$ specifically targets the accuracy of the rendered diffuse color relative to the ground truth as a form of regularization. This distinction is crucial as it addresses an observed trend in joint optimization scenarios where simpler networks often disproportionately capture most of the information. To counteract this bias in training process, we incorporate the $L_{\text{diffuse}}$ loss to ensure that the Diffuse-UNet, which decodes diffuse colors, adequately captures the scene's details. We set $\lambda_{\text{diffuse}} = 0.05$ in our loss function to maintain this balance.

The loss function $L_{\text{normal}}$, which measures the discrepancy between two direction vectors, utilizes cosine similarity to encourage the MLP (described in Section 3.2.2) to predict direction vectors that align closely with the true normals. We set $\lambda_{\text{normal}} = 0.001$, intentionally keeping the weight low. This allows for a slight deviation from the pseudo normal, promoting predictions that are more in line with the physical facts.

Overall, the combination of $L_{\text{renders}}$, $L_{\text{diffuse}}$, and $L_{\text{normal}}$ in the loss function ensures that our method learns light condition and represents the scene more accurately and realistically.

## 4 EXPERIMENTS

### 4.1 Setups

*4.1.1* ***Datasets and Metrics***. We conduct a comprehensive evaluation of our proposed model on 21 scenes from public datasets, including 8 scenes from Shiny Dataset[28], 9 scenes from Mip-NeRF360[2], 2 scenes from Tanks&Temples[11], and 2 scenes in DeepBlending[9]. Consistent with methodologies outlined in previous works[2, 10], we adopted a train/test split approach where every 8th photo was selected for testing. To assess the rendering quality, we further measured the average Peak Signal-to-Noise Ratio (PSNR), Structural Similarity Index Measure (SSIM), and Learned Perceptual Image Patch Similarity (LPIPS)[38].

*4.1.2* ***Implementation Details***. Our method is built upon the widely-used open-source 3D-GS codebase[10]. In line with the approach described in 3D-GS, our models undergo training for 30K iterations across all scenes, employing the same Gaussian density control strategy and training schedule. Furthermore, we have enhanced the differentiable Gaussian rasterization technique to include latent feature and normal vector rendering. All experiments were conducted on an RTX 4090 GPU with 24GB of memory.

### 4.2 Comparisons

We assess the quality of our approach by comparing it to current state-of-the-art baselines, including Instant-NGP[22], Plenoxels[8], Mip-NeRF360[2], 3D-GS[10], Ref-NeRF[28], SpecNeRF[20], Spec-Gaussian[33]. Notably, the latter four methods are specifically designed for 3D scene reconstruction involving specular reflections. The quality metrics for these methods align with the best results reported in their respective published papers. Additionally, we

have re-implemented 3D-GS and Spec-Gaussian on Shiny Dataset, adhering to their hyperparameters. Specifically, Spec-Gaussian is evaluated using anchor Gaussians[19].

*4.2.1* ***Quantitative Comparisons.*** We conducted comparisons on the challenging Shiny Dataset[28], characterized by scenes with stronger specular reflections. As illustrated in Table 1, our approach significantly outperforms all baseline methods on the Shiny Dataset across the metrics mentioned earlier, particularly excelling over models based on 3D-GS. Futhermore, Table 2 presents the results on three real-world datasets[2, 9, 11], highlighting that our approach also achieves competitive performance in rendering quality compared to state-of-the-art methods. These promising results demonstrate the effectiveness of our method in capturing spatial lighting variations under different viewpoints.

**Table 1: Quantitative comparison on Shiny Dataset.**

| Dataset | Shiny Dataset | | |
|---|---|---|---|
| Method\|Metric | PSNR↑ | SSIM↑ | LPIPS↓ |
| Ref-NeRF | 26.50 | 0.724 | 0.283 |
| SpecNeRF | 26.56 | 0.728 | 0.278 |
| 3D-GS | 25.58 | 0.874 | 0.118 |
| Spec-Gaussian | 26.43 | 0.883 | 0.099 |
| Ours | 27.23 | 0.884 | 0.109 |

*4.2.2* ***Qualitative Comparisons.*** Qualitative results across various datasets are provided in Figure 4. On one hand, it is evident that other baselines struggle to accurately model specular highlights or reconstruct correct geometry, whereas our method adeptly captures specular details. On the other hand, under conditions of sparse viewpoints, such as the scene of DrJohnson from the Deep Blending dataset, other models tend to produce artifacts. In contrast, thanks to the Diffuse-UNet, our method is better equipped to capture global image information and complete scene edges. As illustrated in Figure 4, our method clearly delivers smoother rendering results.

### 4.3 Ablation Study

*4.3.1* ***Ablations on different modules***. We have conducted an ablation study on Shiny Dataset to evaluate the effectiveness of using view-mask and Specular-CNN for decoding specular color. The results are shown in Figure 5. We do not fuse the components of specular color for the baseline *Ours no specular* and directly fuse with diffuse color without using mask multiplication when synthesizing specular color for the baseline *Ours no mask*. The former method yields rendering results that closely resemble diffuse reflections, while the latter method achieves better simulation of specular reflection with highligh distributions and shapes that are closer to the original image.

Table 3 further conducts a quantitative analysis that the performance of our method gradually improves upon incorporating the Specular-CNN and viewpoint mask modules. Comparing it to the baseline methods, including the original 3D-GS method (line1) and

Table 2: Quantitative comparison on real-world datasets.

| Dataset | Mipnerf-360 | | | Tanks&Templates | | | Deep Blending | | |
|---|---|---|---|---|---|---|---|---|---|
| Method\|Metric | PSNR↑ | SSIM↑ | LPIPS↓ | PSNR↑ | SSIM↑ | LPIPS↓ | PSNR↑ | SSIM↑ | LPIPS↓ |
| Instant-NGP | 25.59 | 0.699 | 0.331 | 21.72 | 0.723 | 0.178 | 23.62 | 0.797 | 0.423 |
| Plenoxels | 23.08 | 0.626 | 0.463 | 21.08 | 0.719 | 0.379 | 23.06 | 0.795 | 0.510 |
| Mip-NeRF360 | 27.69 | 0.792 | 0.237 | 22.22 | 0.759 | 0.257 | 29.40 | 0.901 | 0.245 |
| 3D-GS | 27.25 | 0.800 | 0.221 | 23.68 | 0.849 | 0.178 | 29.41 | 0.903 | 0.243 |
| Spec-Gaussian | 27.46 | 0.810 | 0.221 | 24.46 | 0.864 | 0.160 | 30.41 | 0.912 | 0.240 |
| Ours | 27.60 | 0.811 | 0.229 | 24.81 | 0.855 | 0.175 | 30.06 | 0.906 | 0.245 |

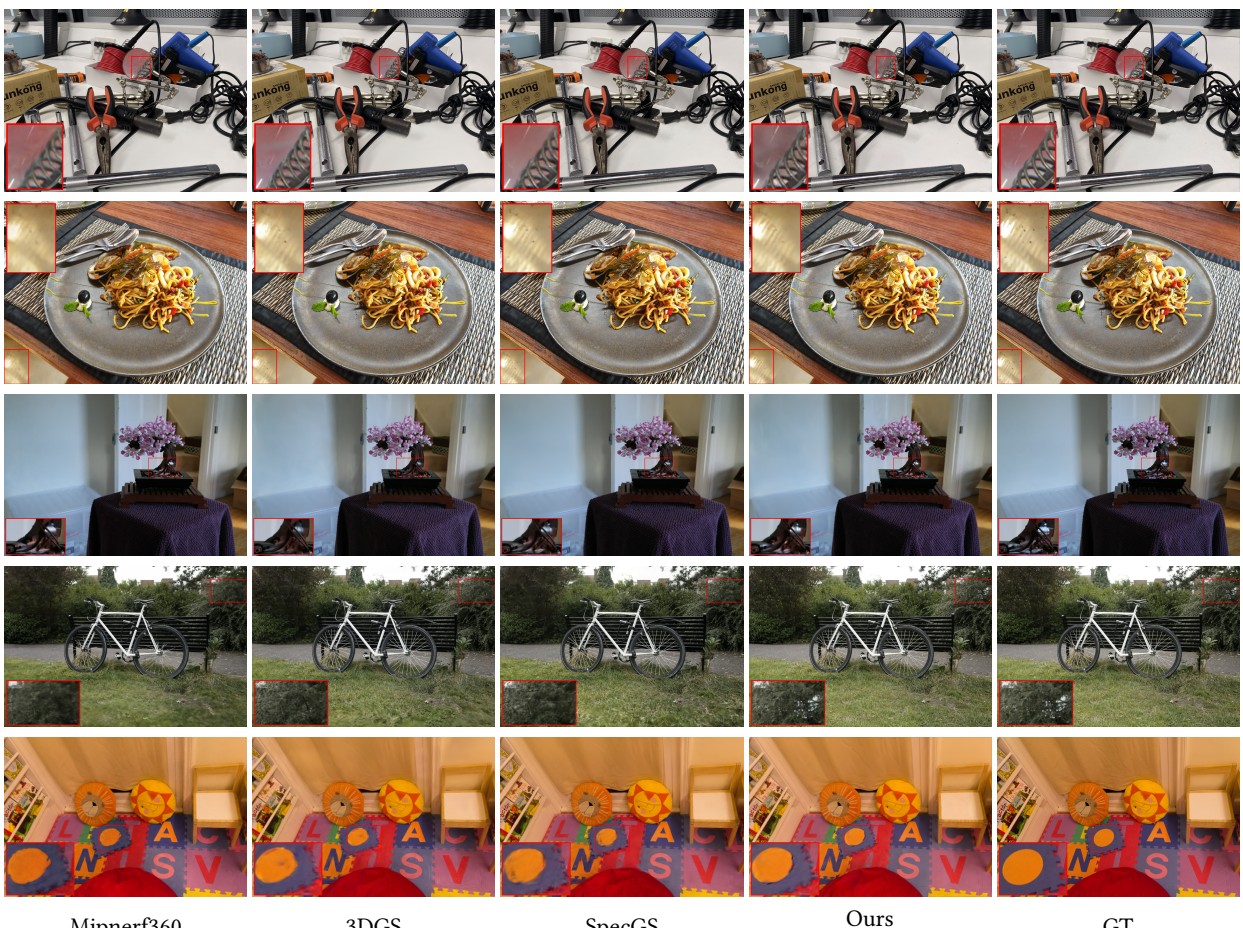

Mipnerf360        3DGS        SpecGS        Ours        GT

Figure 4: Qualitative Comparisons on Shiny Dataset, Mipnerf360 Dataset and Deep Blending Dataset.

the latent-SH approach (line3), the latter does not utilize the view-mask or Specular-CNN for decoding specular color but instead relies on the original 3D-GS's Spherical Harmonic expression for view-dependent color. Our method (line5) outshines the SH-based approach in accurately decoding view-dependent color. Additionally, the latent-SH baseline method also demonstrates superior results compared to the original 3D-GS method, indicating that our latent feature decoding of diffuse color is more effective. In the pre-decode color baseline (line6), we experimented with simplifying the network by placing the color decoding before Gaussian splatting. This involved using an MLP to decode diffuse features and obtain diffuse color, decoding view-dependent features using an MLP after embedding the viewpoint, combining these with the view-mask features to derive specular components, and then merging them directly with the diffuse color. The final rendering was processed through Gaussian splatting. However, this early fusion method underperformed, highlighting the importance of our structured approach.

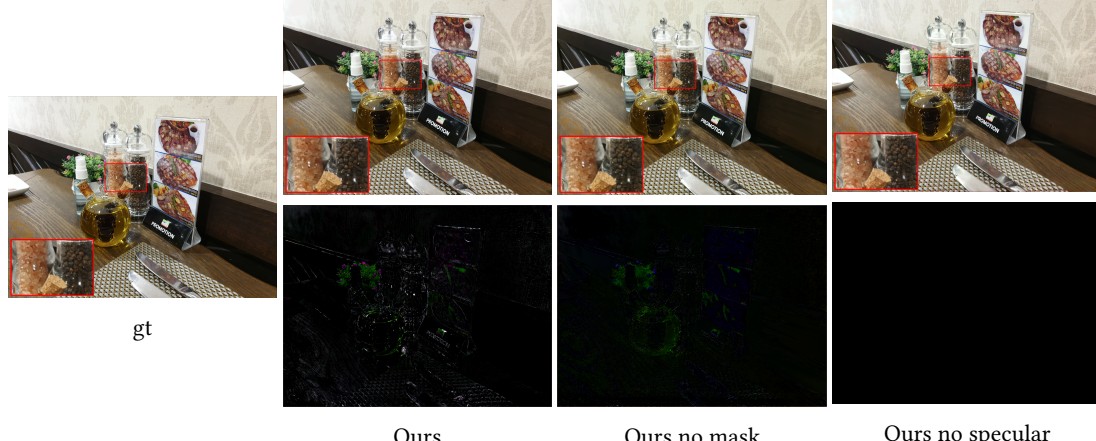

**Figure 5: Ablation on our baseline. We visualize the results of our method after removing the viewpoint mask and removing the specular color components. The top images are the results of our rendering and the bottom images are the orresponding specular color component.**

Through our experiments, we have demonstrated that integrating view-mask and Specular-CNN allows our model to more effectively capture complex lighting distributions and model high-frequency and anisotropic scene features, leading to more precise and realistic rendering results.

**Table 3: Ablations of our method on Shiny Dataset.**

| diffuse color | | specular color | | | Shiny Dataset | | |
|---|---|---|---|---|---|---|---|
| $SH^0$ | latent | $SH^3$ | specular | mask | PSNR↑ | SSIM↑ | LPIPS↓ |
| ✔ | ✘ | ✔ | ✘ | ✘ | 25.58 | 0.874 | 0.118 |
| ✘ | ✔ | ✘ | ✘ | ✘ | 26.53 | 0.876 | 0.122 |
| ✘ | ✔ | ✔ | ✘ | ✘ | 27.04 | 0.878 | 0.113 |
| ✘ | ✔ | ✘ | ✔ | ✘ | 27.05 | 0.882 | 0.111 |
| ✘ | ✔ | ✘ | ✔ | ✔ | 27.23 | 0.884 | 0.109 |
| pre-decode color | | | | | 26.17 | 0.880 | 0.111 |

*4.3.2* ***Ablations on different channels***. To investigate the impact of the number of channels on our model, we conducted experiments with different channel configurations. Typically, our method employs a configuration of 8 + 8 latent feature channels, where 8 dimensions are allocated for latent diffuse features and another 8 for latent view-dependent features. We evaluated the performance with varying these numbers to 4 + 4 and 16 + 16. The outcomes, detailed in Table 4, demonstrate that models with more channels generally perform better. However, the improvement between the 8 + 8 and 16 + 16 configurations is marginal. Considering this and the balance between computational efficiency and effectiveness, we opted for the 8 + 8 channel configuration, which optimizes the trade-off between performance and training time, ensuring efficient yet effective rendering.

**Table 4: Ablations about the impact of varying the number of latent feature channels on the Shiny Dataset.**

| scene | channels diffuse+spec | PSNR↑ | SSIM↑ | LPIPS↓ |
|---|---|---|---|---|
| food | 4 + 4 | 23.30 | 0.808 | 0.160 |
| | 8 + 8 | 23.55 | 0.819 | 0.140 |
| | 16 + 16 | 23.71 | 0.820 | 0.128 |
| seasoning | 4 + 4 | 28.87 | 0.906 | 0.105 |
| | 8 + 8 | 29.49 | 0.913 | 0.102 |
| | 16 + 16 | 29.21 | 0.914 | 0.089 |
| average | 4 + 4 | 26.09 | 0.857 | 0.133 |
| | **8 + 8 (Ours)** | 26.52 | 0.866 | 0.121 |
| | 16 + 16 | 26.46 | 0.867 | 0.108 |

## 5 CONCLUSION

In this paper, we introduce a novel method called Lantent-SpecGS that aims to improve the modeling of specular reflections and handle anisotropic appearance components in 3D Gaussian Splatting (3D-GS). The proposed method achieves more efficient representation of 3D feature fields, including geometry and appearance, by utilizing universal latent neural descriptors within each 3D Gaussian . To decode the splatting feature maps into diffuse and specular colors, the method employs two parallel convolutional neural networks. The final rendered image is obtained by merging these two colors using a viewpoint-dependent mask learned during training . Experimental results demonstrate the competitiveness of the proposed method in novel view synthesis and its enhanced capability to handle complex scenes with specular reflections. In future work, we will explore the consistent representation of cross-modal neural descriptors to achieve multi-modal tasks using latent features.

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
