# OpenReview forum: "SpecGaussian with latent features: A high-quality modeling of the view-dependent appearance for 3D Gaussian Splatting"
_acmmm.org/ACMMM/2024/Conference — MM2024 Poster_

### Official Review · Reviewer_dbzp · 2024-04-29

**Rating:** 5
**Confidence:** 3

**Summary:**

This paper introduces a new 3D-GS based on latent feature and two parallel decoder.

**Strengths:**

This paper is technically novel. The two main innovations are good and easy to follow.

**Limitations:**

In the 3-rd paragraph of the Introduction, the author declare that the SH coefficients used in 3D-GS lead to excessive memory usage, and SH "does not resolve the issue of significant artifacts that frequently occur within 3D-GS". How much memory can you save with latent feature, do you have experimental results or mathematical analysis? Are there any cases that 3D-GS show artifacts but your method does not?

**Suitability:**

2

---

### Official Review · Reviewer_ssWm · 2024-05-15

**Rating:** 3
**Confidence:** 4

**Summary:**

This paper addresses the issues of modeling view-dependent effects under complex lighting conditions in a 3D-GS based model. It introduces a latent neural descriptor to effectively represent the features of both appearance and geometry of each Gaussian feature. Additionally, it designs a feature decoder to decouple diffuse color and specular color, allowing for more flexibility in the final rendering process. By separating these two components, the model can better handle the complex interactions caused by variant viewpoints, resulting in more accurate and visually pleasing rendered images.

**Strengths:**

1. This paper is somewhat innovative. It proposes a latent neural descriptor to represent each 3D Gaussian feature, allowing for a more compact representation of the scene while retaining view-dependent effects.
2. This paper is technically feasible. The 3D-GS method often overlooks the underlying structure of the scenes, which can result in suboptimal representations. The splitting and pruning strategies may introduce redundancies or omissions in the Gaussian points, leading to reduced robustness when handling object edges or scene boundaries. The Diffuse-UNet network may address these issues by refining rendered images.

**Limitations:**

1. This paper achieves slight improvement and lacks comparison with the state-of-the-art methods. For example, in the case of the shiny dataset, notable advancements can be found in works such as "Boosting View Synthesis with Residual Transfer" and "Light Field Neural Rendering." Similarly, for real-world datasets, significant progress has been made in approaches such as "Scaffold-GS: Structured 3D Gaussians for View-Adaptive Rendering."
2. This paper focuses solely on the impact on rendering quality while neglecting rendering speed. However, one of the key expectations of 3D-GS methods is real-time rendering.
3. The ablation experiments demonstrate that "removing the viewpoint mask" and "removing the specular color components" exhibit some level of decoupling, but in Figure 5, the rendered images do not show significant differences or noticeable effects.

**Suitability:**

3

---

### Official Review · Reviewer_x8vp · 2024-05-25

**Rating:** 4
**Confidence:** 3

**Summary:**

This paper develops a novel latent 3D Gaussian splatting method to achieve high-quality rendering. The authors utilize latent features instead of spherical harmonics (SH) to indirectly produce color through diffuse and specular decoders. Although the method does not achieve state-of-the-art performance, the experiments demonstrate its effectiveness.

**Strengths:**

The idea is novel and contributions are enough.
They clearly present the idea on the paper
They demonstrate their results on both shiny results and real-world dataset.

**Limitations:**

The connection between view mask features and BRDF function is not very concrete and theoretically reasonable.
More results of the color decomposition  are needed to show in the paper
The results on three real world dataset are not very significant. Spec_Nerf is not compared on the real world dataset
The efficiency of the proposed method should be presented and compared with other method.
The proposed method claims the effectiveness under sparse view conditions should be better validated

**Suitability:**

3

---

### Official Review · Reviewer_Jv8V · 2024-05-25

**Rating:** 4
**Confidence:** 1

**Summary:**

This paper proposes a 3D Gaussian splatting framework aiming to improve specular reflection rendering.

**Strengths:**

1. This paper extracts additional informations from latent features attached to the 3DGS and uses them to better enhance the specular reflection rendering;
2. Provides quite sufficient experiments with sota results in shiny senarios;

**Limitations:**

1. This paper mainly compares with Spec-Gaussian, but does not show better results in the real world scenario:
     a. Why are the results less effective than other models in some scenarios and categories? Is it due to the introduction of spectral-related latents?
     b. If there are advantages in other aspects such as memory consumption or runtime, it is recommended to include them in the Experiments of main paper.

**Suitability:**

3

---

### Meta-Review · Area_Chair_eQYg · 2024-06-27

**Recommendation:** Accept (Poster)
**Confidence:** 4

**Metareview:**

The paper initially received 1 Weak Accept, 2 Borderline Accept, and 1 Borderline Reject. After rebuttal, one reviewer downgraded the score from Weak Accept to Weak Reject.

The contribution of this paper is acknowledged by the reviewers, including the two reviewers on the negative side. The major concerns of the reviewers lie in the efficiency as well as rendering quality. After checking the paper and rebuttal, AC believes that the rendering efficiency, though reduced compared to the original GS, is reasonable considering the heightened difficulty of problem. It is also believed that the diminished advantage on real-world datasets is reasonable since the method focuses on handling highlights which may not pervade the scenes in these datasets.

The AC recommends accepting the paper since its contributions outweighed its shortcomings. The authors should carefully revise the paper according to the reviewer's comments and suggestions.